

# Diversity of fall armyworm, *Spodoptera frugiperda* and their gut bacterial community in Kenya

Joseph Gichuhi[1], Subramanian Sevgan[1], Fathiya Khamis[1],
Johnnie Van den Berg[2], Hannalene du Plessis[2], Sunday Ekesi[1] and
Jeremy K. Herren[1,3]

[1] International Centre of Insect Physiology and Ecology (ICIPE), Nairobi, Kenya
[2] Unit for Environmental Sciences and Management, North-West University, Potchefstroom, South Africa
[3] MRC-University of Glasgow Centre for Virus Research, Henry Wellcome Building, Glasgow, UK

## ABSTRACT

**Background:** The invasive fall armyworm, *Spodoptera frugiperda* (J.E. Smith) is a polyphagous pest that causes widespread damage particularly to maize and sorghum in Africa. The microbiome associated with *S. frugiperda* could play a role in the insects' success and adaptability. However, bacterial communities in *S. frugiperda* remain poorly studied.
**Methods:** We investigated the composition, abundance and diversity of microbiomes associated with larval and adult specimens of *S. frugiperda* collected from four maize growing regions in Kenya through high throughput sequencing of the bacterial 16S rRNA gene. The population structure of *S. frugiperda* in Kenya was assessed through amplification of the mitochondrial cytochrome oxidase subunit I gene.
**Results:** We identified Proteobacteria and Firmicutes as the most dominant bacterial phyla and lesser proportions of Bacteroidetes and Actinobacteria. We also observed differences in bacterial microbiome diversity between larvae and adults that are a likely indication that some prominent larval bacterial groups are lost during metamorphosis. However, several bacterial groups were found in both adults and larvae suggesting that they are transmitted across developmental stages. Reads corresponding to several known entomopathogenic bacterial clades as well as the fungal entomopathogen, *Metarhizium rileyi*, were observed. Mitochondrial DNA haplotyping of the *S. frugiperda* population in Kenya indicated the presence of both "Rice" and "Corn" strains, with a higher prevalence of the "Rice" strain.

## INTRODUCTION

Invasions by exotic pests can have major detrimental effects on agricultural production and natural resources (*Huber et al., 2002*). The fall armyworm (FAW), *Spodoptera frugiperda* (J.E. Smith) (Lepidoptera: Noctuidae) is a polyphagous pest that is, native to tropical regions of the western hemisphere, where it is known for its ability to cause economic damage to several crop species. In 2016, *S. frugiperda* was first detected in

Corresponding author
Jeremy K. Herren, jherren@icipe.org

West Africa (*Goergen et al., 2016*), and since then this pest has rapidly spread across the continent (*Day et al., 2017*; *Nagoshi et al., 2018*; *Rwomushana et al., 2018*). By 2018, *S. frugiperda* was reported in all countries in Sub-Saharan Africa except Djibouti and Lesotho (*Rwomushana et al., 2018*). Furthermore, *S. frugiperda* also has now reached the continent of Asia (*Deole & Paul, 2018*; *Sisodiya et al., 2018*). Maize and other economically important food crops in these regions are extensively damaged by *S. frugiperda* larvae (*Day et al., 2017*) causing extensive economic losses and threatening food security. Genetic characterizations have supported the initial postulation that this pest species exists in two subpopulations called the "Rice" and "Corn" strains that are preferentially associated either with smaller grasses (such as rice and bermudagrass) or with larger grasses (such as sorghum and maize) respectively (*Nagoshi & Meagher, 2004*). Strain composition in populations of this pest may therefore have ramifications on the variety of crops at risk of infestation (*Nagoshi et al., 2019*).

There is a lack of information about *S. frugiperda*-host plant interactions and other factors that may be leading to the rapid spread of *S. frugiperda* in the geographic regions that have recently been invaded. Many of the control measures used in the western hemisphere (e.g., transgenic maize, chemical insecticides) might not be readily available and economically viable for subsistence farmers in Africa. Furthermore, the use of highly hazardous pesticides is not considered a sustainable long term control measure for any pest (*FAO, 2018*). In addition, *S. frugiperda* have been reported to evolve resistance to most chemical insecticides (e.g., pyrethroids, organophosphates and carbamates) (*Yu, 1991*) and to transgenic maize that are used in its control (*Jakka et al., 2016*; *Banerjee et al., 2017*; *Flagel et al., 2018*; *Botha et al., 2019*). As a consequence, there is a great need for alternative, cost-effective control strategies for *S. frugiperda* (*FAO, 2018*).

A recent survey in Ethiopia, Kenya and Tanzania indicated that *S. frugiperda* has established interactions with indigenous parasitoid species (*Sisay et al., 2018*) that could be harnessed for biological control. A study on *S. frugiperda* host plant interactions in East Africa has also suggested a climate adapted push–pull system (*Midega et al., 2018*) and maize–legume intercropping (*Hailu et al., 2018*) for management of pests including FAW on maize farms. However, many factors related to *S. frugiperda* rapid spread, host plant interactions, bio-ecology and insect-microbiome interactions in the African region remain poorly understood.

Insect microbiomes can have important consequences for the outcome of insect pest-natural enemies- host plant interactions (*Ferrari, Vavre & Lyon, 2011*). Strategies that involve modifying insect microbiomes are currently being evaluated for control and management of pests and vectors of plant diseases (*Crotti et al., 2012*; *Perilla-Henao & Casteel, 2016*; *Arora & Douglas, 2017*; *Beck & Vannette, 2017*). Insect microbiomes play a key role in the adaptation of insects to their environment and are therefore a major and often poorly understood determinant of the host plant and geographic range of insect pests (*Su, Zhou & Zhang, 2013*). In general, a greater diversity of microbial symbionts exist within the insect's gut lumen, while few others exist inside cells of the host, or on the cuticle (*Douglas, 2015*). Gut microbial symbionts are known to influence their host's nutrition, usually by promoting digestion and availability of nutrients (*Douglas, 2009*).

These symbionts can also modulate the immune response and accessibility of the host to invading organisms, and therefore have direct or indirect effects on host susceptibility to parasites and pathogens (*Garcia et al., 2010*; *McLean & Godfray, 2015*; *Ubeda, Djukovic & Isaac, 2017*). Previous studies have also identified important roles of bacterial symbionts in the interactions between phytophagous insects and host plants (*Frago, Dicke & Godfray, 2012*; *Biere & Bennett, 2013*; *Brady & White, 2013*). In addition, microbial symbionts can break down complex molecules such as insecticides and promote insecticide resistance (*Kikuchi et al., 2012*; *Xia et al., 2018*). It is also notable that pathogenic bacteria can reside in host guts, only initiating or facilitating pathogenesis under certain conditions (*Wei et al., 2017*). Studying the gut microbiome is not only important from the standpoint of understanding mutualistic relationships but also for laying the foundation for future projects aimed at developing microbial biocontrol agents.

There are an increasing number of studies examining the microbial diversity of lepidopterans. While in some of the assessed species consistent bacterial communities have been observed in both field and laboratory collected populations as well as in insects reared on different diets (*Broderick et al., 2004*; *Xiang et al., 2006*; *Pinto-Tomás et al., 2011*), other studies reported no host specific resident communities that occurred, regardless of the insect diet (*Hammer et al., 2017*). It is possible that lepidopterans are less prone to forming robust "core" microbiomes due to several factors: (1) very high pH in the midgut, (2) low retention time of food, (3) lack of microbe housing structures in the intestinal tract, and (4) continual replacement of the peritrophic matrix (*Hammer et al., 2017*). Nevertheless, bacterial communities do continually associate with lepidopterans and influence a variety of important host processes (*Broderick, Raffa & Handelsman, 2006*; *Anand et al., 2010*; *Wang et al., 2017*).

Relatively few studies have assessed the *Spodoptera*-associated gut microbiome. In a recent study, the microbial diversity of *Spodoptera exigua* (Hübner) (Lepidoptera: Noctuidae) was examined by 16S rDNA sequence profiling (*Gao et al., 2019*). In *Spodoptera exigua*, the dominant bacterial clades are Proteobacteria and Firmicutes, with the predominant genus in larvae being *Enterococcus*. In *S. frugiperda*, previous studies have isolated several bacterial strains using culture-dependent methods (*De Almeida et al., 2017*; *Acevedo et al., 2017*).

In this study, we used 16S rDNA sequence profiling to characterize the diversity of bacteria associated with populations of *S. frugiperda* in Kenya and assessed the prevalence of the *S. frugiperda* strains in these populations using mitochondrial COI gene sequences. Specifically, we characterized the structure of the circulating *S. frugiperda* populations in Kenya as well as the gut bacterial communities derived from both larval and adult specimens collected in different agro-ecological zones. Understanding pest population structures is important for understanding invasion patterns and planning with regards to strain-specific susceptibility of crops, whereas characterizing pest-associated microbiomes is a useful foundation for exploring insect-microbiome interactions that could be exploited to improve control strategies.

## MATERIALS AND METHODS

### Insect collection

*Spodoptera frugiperda* larvae were collected from infested maize fields in Kenya between June and December 2017 at the following locations: Ngeria (N00.37024 E035.9862) and Burnt Forest (N00.22505 E035.42479) in Uasin Gishu County; Msamia, Kitale (N00.98009 E034.97170) in Trans Nzoia County; Shimba Hills (S04.33228 E039.34361) in Kwale County and Chala Irrigation Scheme (S03.27338 E037.13816) and Wundanyi (S03.337538 E038.33612) in Taita Taveta County. Part of the field collected insects from each sampled region in Kenya were reared on fresh maize leaves in ventilated cages to pupation and eclosion at 27 °C and 60% humidity, while the rest were stored in absolute ethanol at −20 °C. We profiled the bacterial microbiome for 18 samples from four of these locations, whereas we included samples from all the sampled locations for mtDNA haplotyping (Fig. 1).

### DNA extraction and 16S rDNA sequencing

Guts from nine live stage 5–6 larvae and nine 1 day old emerging adults from the Kenya collected samples were dissected separately in phosphate buffered saline (PBS) following surface sterilization and used for DNA extraction. Insects were surface sterilized in 70% ethanol, in 5% v/v sodium hypochlorite solution followed by three washes in PBS for 3 min in each solution. Each dissected gut tissue was homogenized in PBS using five 4 mm diameter ceramic beads in a two ml microfuge tube, using a TissueLyser II beadmill (Qiagen, Hilden, Germany). DNA was extracted using the ISOLATE II Genomic DNA Kit (Bioline, London, UK) according to the manufacturer's instructions. DNA extracted from gut samples was submitted for high throughput sequencing targeting the v4 region of the bacterial 16s rRNA gene using the Illumina Miseq platform (Center for Integrated Genomics, University of Lausanne, Switzerland). Sequence reads were checked for quality using FastQC v 0.11.28 (*Andrews, 2010*) and pre-processed to remove adapters and sequencing primers using Cutadapt v1.18 (*Martin, 2011*). Forward and reverse reads were imported into QIIME2-2018.11 (*Boylen et al., 2018*). The deblur plugin (*Amir et al., 2017*) was used to further filter the reads based on per base quality scores and merge the paired-end-reads and cluster reads into operational taxonomic units (OTUs). Taxonomic assignment was done using the blast classifier against the Silva132 reference database (*Quast et al., 2013*) at a 99% identity cutoff. OTU prevalence and variance based filtering as well as alpha and beta diversity measures were applied to the data in the Microbial Analyst Marker Data Profiling (*Dhariwal et al., 2017*). Shannon diversity indices were applied along with Mann–Whitney and analysis of variance statistics in profiling alpha diversity between sets of samples. Beta diversity was evaluated using Bray–Curtis and unweighted Unifrac distances. Significance testing was done using permutational multivariate analysis of variance (PERMANOVA) and visualization done through non-metric multidimensional scaling (NMDS) ordination. The empirical analysis of digital gene expression data in R (edgeR) algorithm (*Robinson, McCarthy & Smyth, 2009*) was used to evaluate differential abundance of bacterial genera reads between sample groups.

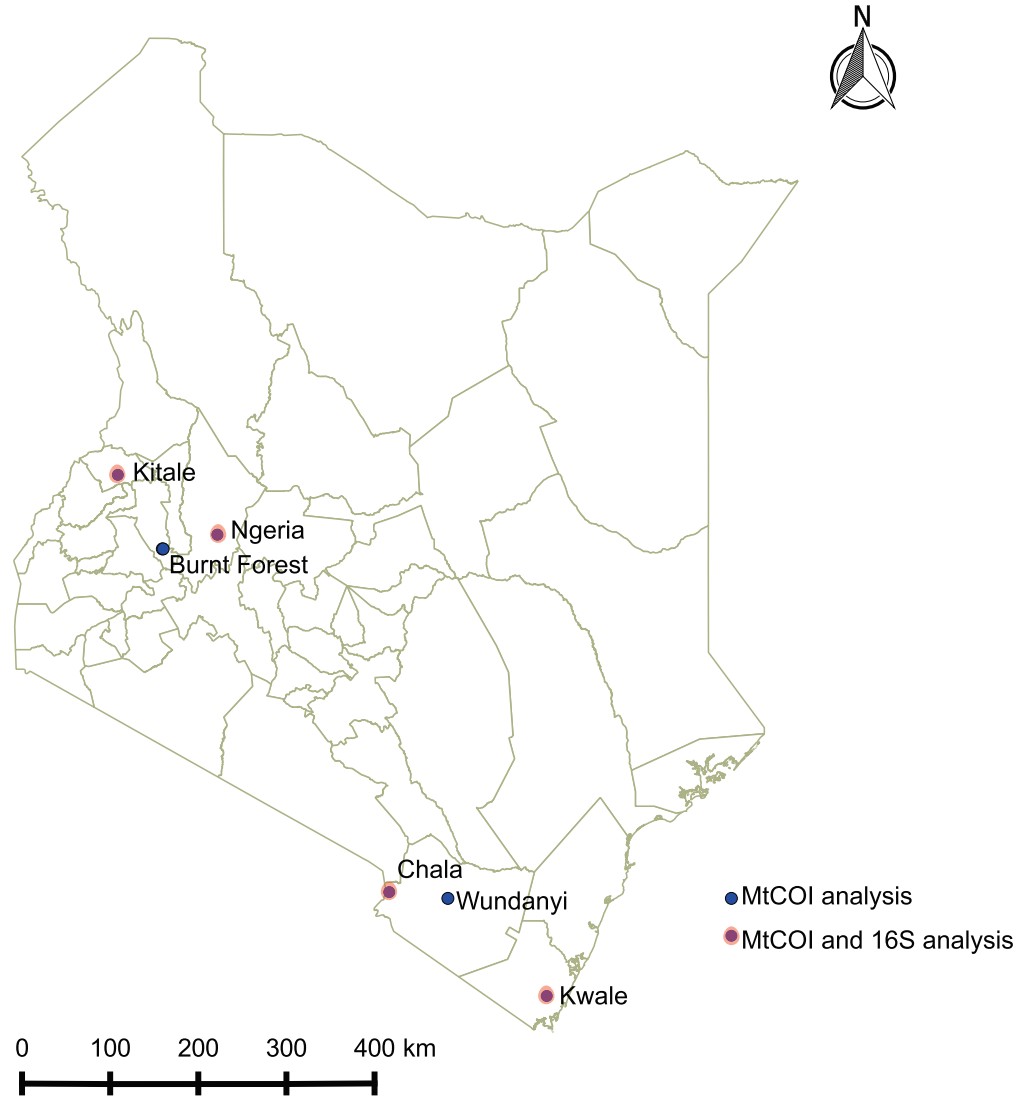

**Figure 1** Sites from which *Spodoptera frugiperda* larvae were collected in Kenya.

All sequence reads were archived in the Sequence Read Archive under the BioProject: PRJNA521837.

## mtDNA haplotyping

DNA was extracted from surface-sterilized whole insects using the ISOLATE II Genomic DNA Kit (Bioline, London, UK) according to the manufacturer's instructions. Mitochondrial COI gene sequences were amplified from insect DNA by PCR using the primer LCO1490 and HCO2198 (*Folmer et al., 1994*). Reactions were set up in total volumes of 10 µl each, containing 5× MyTaq reaction buffer (5 mM dNTPs, 15 mM MgCl2, stabilizers and enhancers) (Bioline, London, UK), 2 µM of each primer, 0.25 mM MgCl2 (Thermo Fischer Scientific, Waltham, MA, USA), 0.125 µl MyTaq DNA

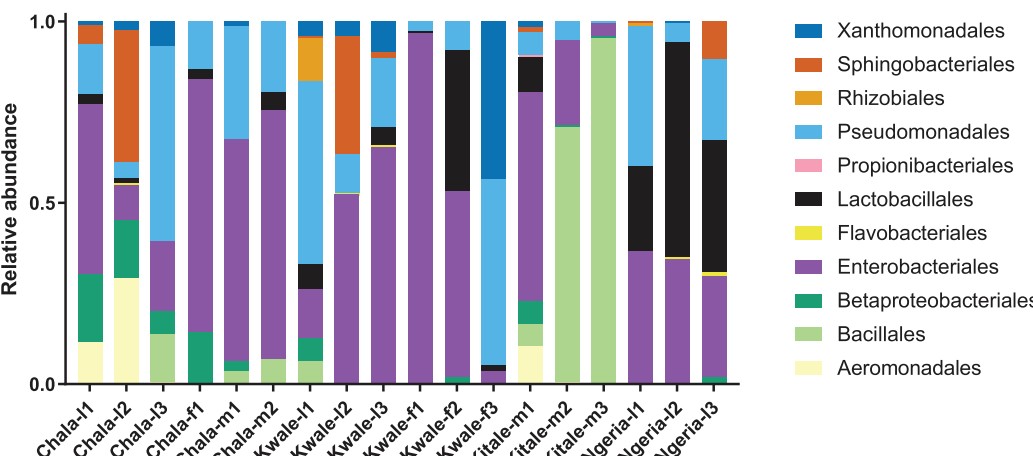

**Figure 2 Composition of bacterial OTUs at Order level in screened *Spodoptera frugiperda* samples.** Samples are denoted as "site name-developmental stage/sex_sample number", where: l, larvae; m, male and f, female.

polymerase (Bioline, London, UK), and 7.5 ng/µl of DNA template. These reactions were set up in a Master cycler Nexus gradient thermo-cycler (Thermo Fischer Scientific, Waltham, MA, USA) using the following cycling conditions: initial denaturation for 2 min at 95 °C, followed by 40 cycles of 30 s at 95 °C, 45 s at 50.6 °C and 1 min at 72 °C, then a final elongation step of 10 min at 72 °C. PCR products were separated by 1% agarose gel electrophoresis and visualized by ethidium bromide staining and UV trans-illumination. Direct sequencing was done for all host mtCOI gene and the sequences deposited in the GenBank.

## RESULTS

A total of 457,501 sequence reads were retained after removal of spurious reads and all reads shorter than 220, which was the median length of all sequences with a quality score higher than 20. These sequences clustered into 1,796 OTUs. Of these, 197 OTUs survived low count and interquartile range-based variance filtering to eliminate OTUs that could arise from sequencing errors and contamination. OTUs initially characterized as "*Candidatus hamiltonella*" by comparison to the Silva132 reference database were re-analyzed by homology searches against the NCBI nr nucleotide database through blast (*Altschul et al., 1990*) and found to be *Pseudomonas*, highlighting a potential incorrect assignment in the reference database.

The most abundant bacterial Phyla observed across the FAW gut samples were Proteobacteria, Firmicutes, Bacteroidetes and a small proportion of Actinobacteria (Fig. S1). OTUs clustering in the orders Enterobacteriales and Pseudomonadales were predominant in the majority of the samples (Fig. 2).

We noticed that despite the high genus-level diversity between samples (Fig. 3), there were some similarities based on developmental stage and location. For example, there was a very high proportion of: (1) *Pseudomonas* in the two adult male samples from Chala, (2) *Citrobacter* in two larval samples from Kwale, (3) *Lysinibacillus* in two male samples

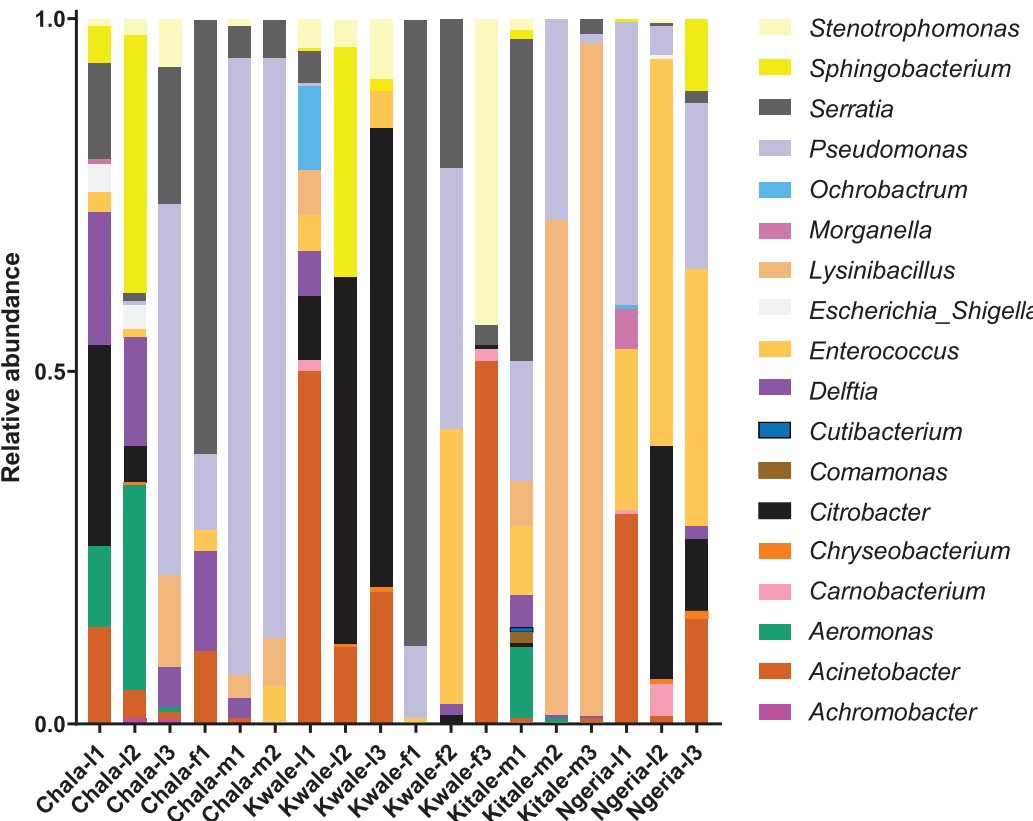

**Figure 3 Genus level composition of (% of OTUs) in the different samples of *S. frugiperda*.** Samples are denoted as "site name-developmental stage/sex_sample number", where: l, larvae; m, male and f, female. The relative abundances of the 19 most abundant genera (with more than 10 counts after low count and variance filtering) are represented here.

from Kitale and (4) *Enterococcus* in two larval samples from Ngeria. It was noted that *Stenotrophomonas*, *Sphingobacterium*, *Serratia*, *Pseudomonas*, *Morganella*, *Enterococcus* and *Delftia* were detected in both larvae and adult samples.

In one of the larval samples from the Ngeria site (Ngeria-l2), we observed an excessive number of non-bacterial reads. Through homology searches against the NCBI nr nucleotide database, these were found to be closely related to *Metarhizium rileyi* (Farl.) *Kepler et al. (2014)* (formerly *Nomuraea rileyi*), an entomopathogenic fungus that is, known to infect *S. frugiperda* (Fig. 4).

The bacterial OTU richness appeared to be higher in *S. frugiperda* larvae than adults, however this difference was not statistically significant (*p*-value: 0.062526; (Mann–Whitney) statistic: 19) using Shannon diversity metrics (Fig. 5A). In addition, no significant variation in OTU richness and abundance was observed between larvae from different sampling sites (*p*-value: 0.32834; (ANOVA) *F*-value: 1.3486) (Fig. 5B).

The composition of bacterial OTUs between larvae and adult *S. frugiperda* was observed to overlap although some significant dissimilarity ((PERMANOVA) *F*-value: 2.734; *R*-squared: 0.26715; *p*-value < 0.001 (NMDS) Stress = 0.13859) was recorded (Fig. 6). Similarly, OTU composition was observed to vary significantly among larval samples from

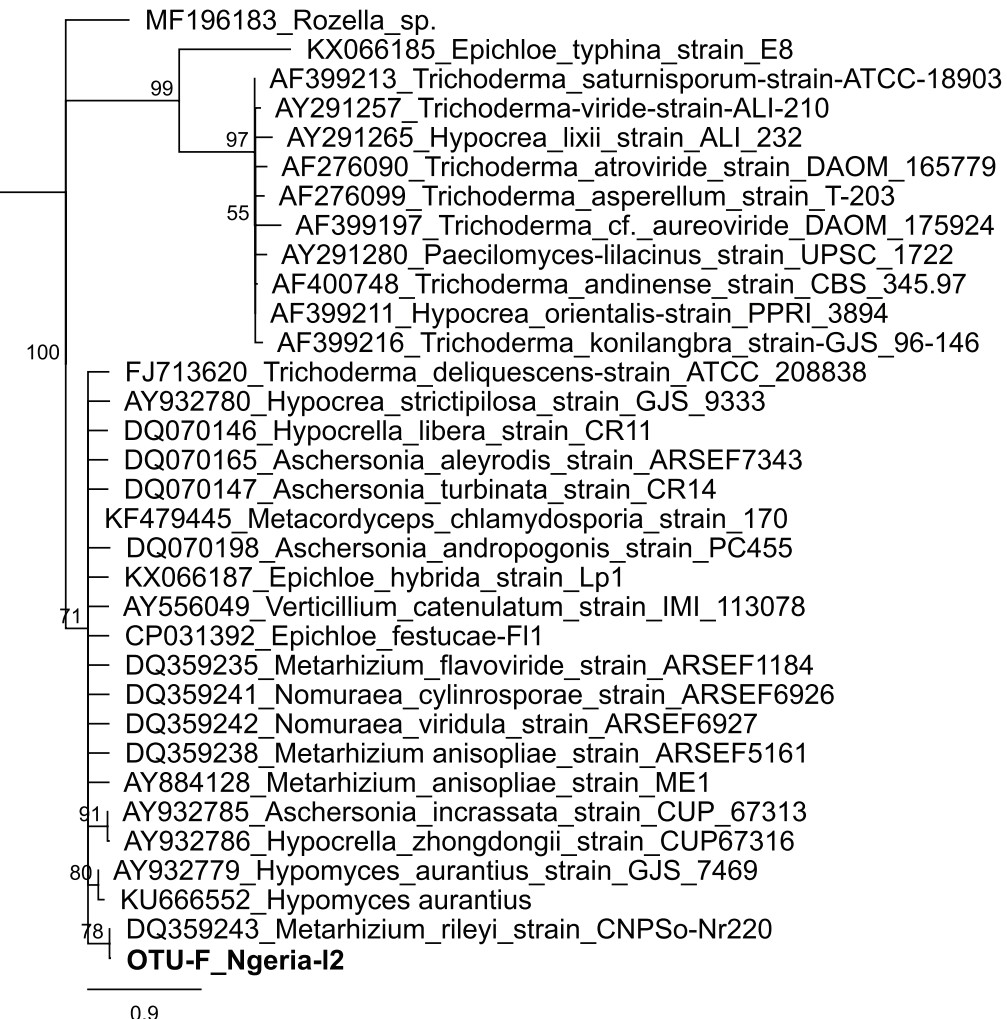

**Figure 4 Neighbor-joining tree of fungus OTU detected in *S. frugiperda* sample (Ngeria-l2; in bold) and GenBank accessions of small subunit ribosomal RNA gene sequences from related fungi.** Sequences are labeled by their GenBank accessions followed by genus, species and strain where available. Bootstrap values are indicated above the branches. Branches with a bootstrap value less than 50 are collapsed. A sequence from a species in the genus *Rozella* is included as an out-group.

different sites ((PERMANOVA) *F*-value: 1.7511; *R*-squared: 0.36856; *p*-value < 0.037 (NMDS) Stress = 0.057109) (Fig. 7).

A significant differential abundance was observed for three bacterial genera between larvae and adult *S. frugiperda* samples using the EdgeR algorithm at an adjusted *p*-value of 0.05. Two of these: *Citrobacter* (log2FC = 4.4178, *p* value = 3.6E−6, FDR = 7.218E−5) and *Sphingobacterium* (log2FC = 3.625, *p* value = 1.01E−4, FDR = 0.0010118) were more abundant in larvae whereas the third: *Lysinibacillus* (log2FC = −3.2247, *p* value = 4.4E−3, FDR = 0.029375) was more abundant in adults (Fig. 8).

Based on mtDNA sequences, the *S. frugiperda* strains detected in this study were identical to strains from Canada, USA and Brazil, as well as strains that were recently

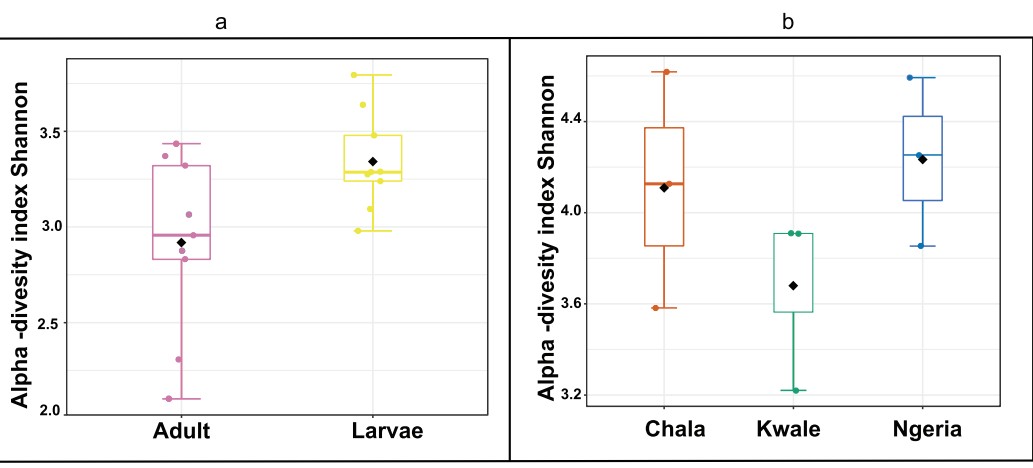

(p-value: 0.062526; [Mann-Whitney] statistic: 19)     (p-value: 0.32834; [ANOVA] F-value: 1.3486)

**Figure 5 A comparison of the Shannon diversity indices for: (A) adult and larval samples from all sites and (B) larvae collected from different sites.** The Shannon diversity index (H') was calculated based on the OTU-level of classification. The boxplots show the distribution of H' values across all samples.               

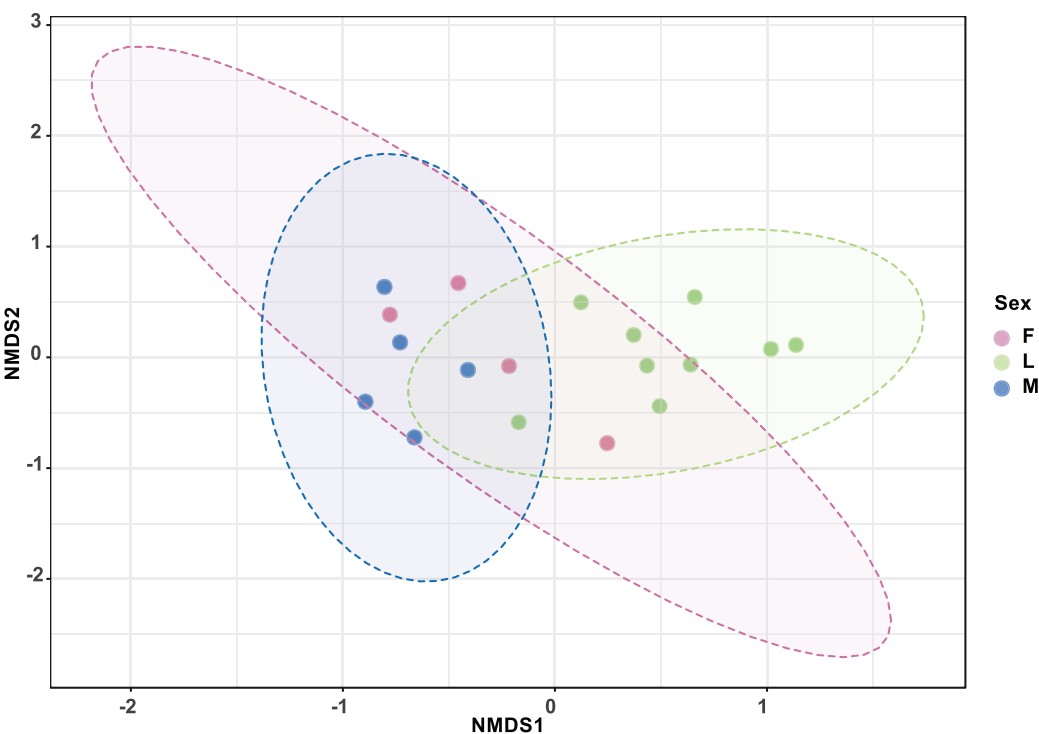

[PERMANOVA] F-value: 2.734; R-squared: 0.26715; p-value = 0.001 [NMDS] Stress = 0.13859

**Figure 6 Non-metric multidimensional scaling (NMDS) ordination based on Bray–Curtis dissimilarities in bacterial communities detected in the *S. frugiperda* samples.** Samples are colored according to their developmental stage and sex as indicated on the legend where: F, female; L, larvae and M, male.               

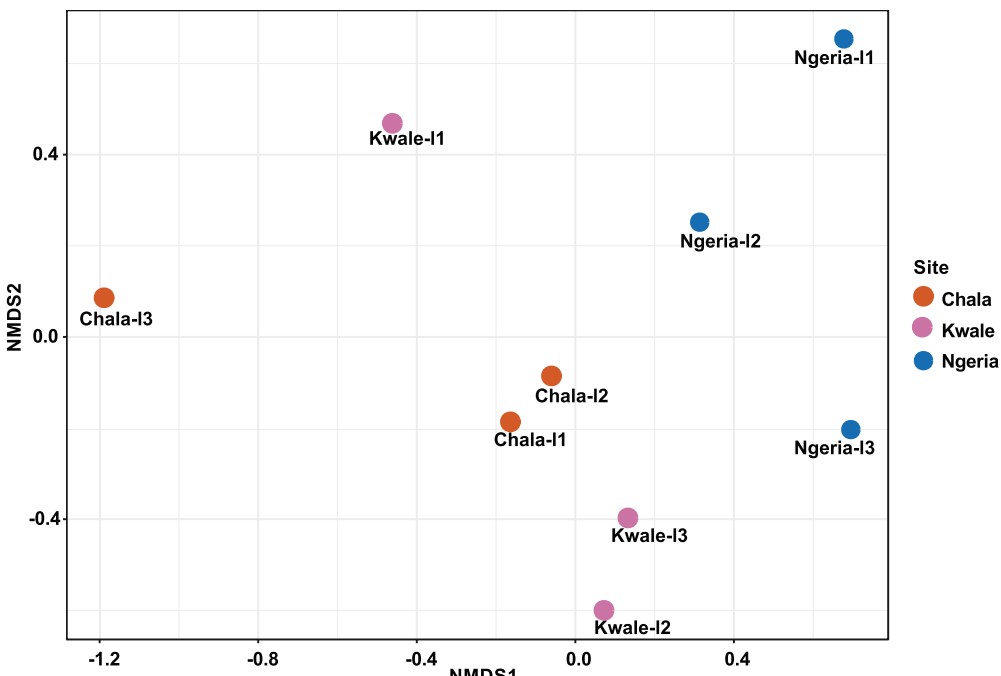

[PERMANOVA] F-value: 1.7511; R-squared: 0.36856; p-value = 0.037 [NMDS] Stress = 0.057109

**Figure 7 Non-metric multidimensional scaling (NMDS) ordination based on Weighted Unifrac distance dissimilarities in bacterial OTU composition between larval sample pairs from different sites.** Samples are labeled by their collection site identifiers.

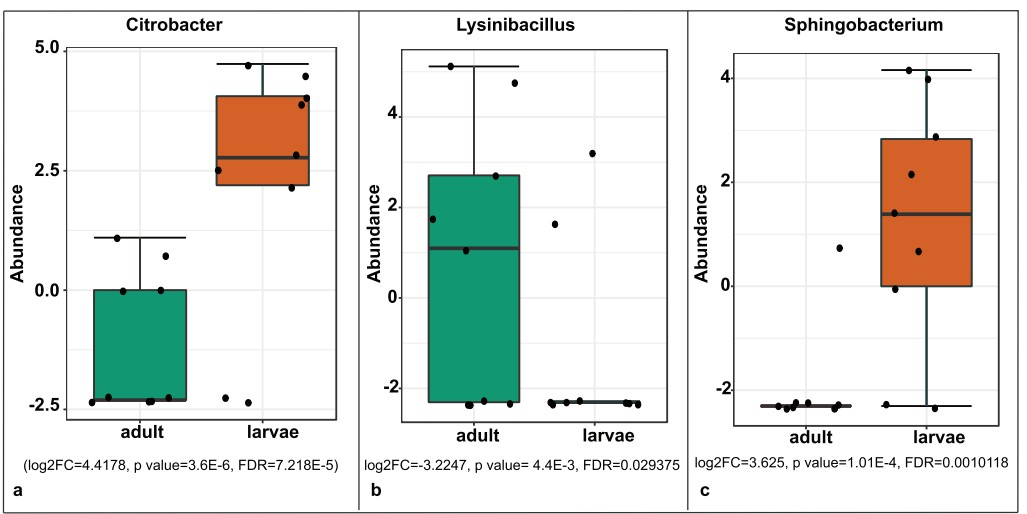

**Figure 8 Comparative abundance of (A) *Citrobacter*, (B) *Lysinibacillus* and (C) *Sphingobacterium* between adults and larvae of *S. frugiperda*.** Abundance is shown on a log transformed scale of original counts.

reported in Kenya and parts of Africa and India (Fig. 9). All the samples clustered in two major clades widely referred to as either the "Rice" or the "Corn" strain (hereafter referred to as R-strain and C-strain). We investigated the frequency of mtDNA haplotypes of

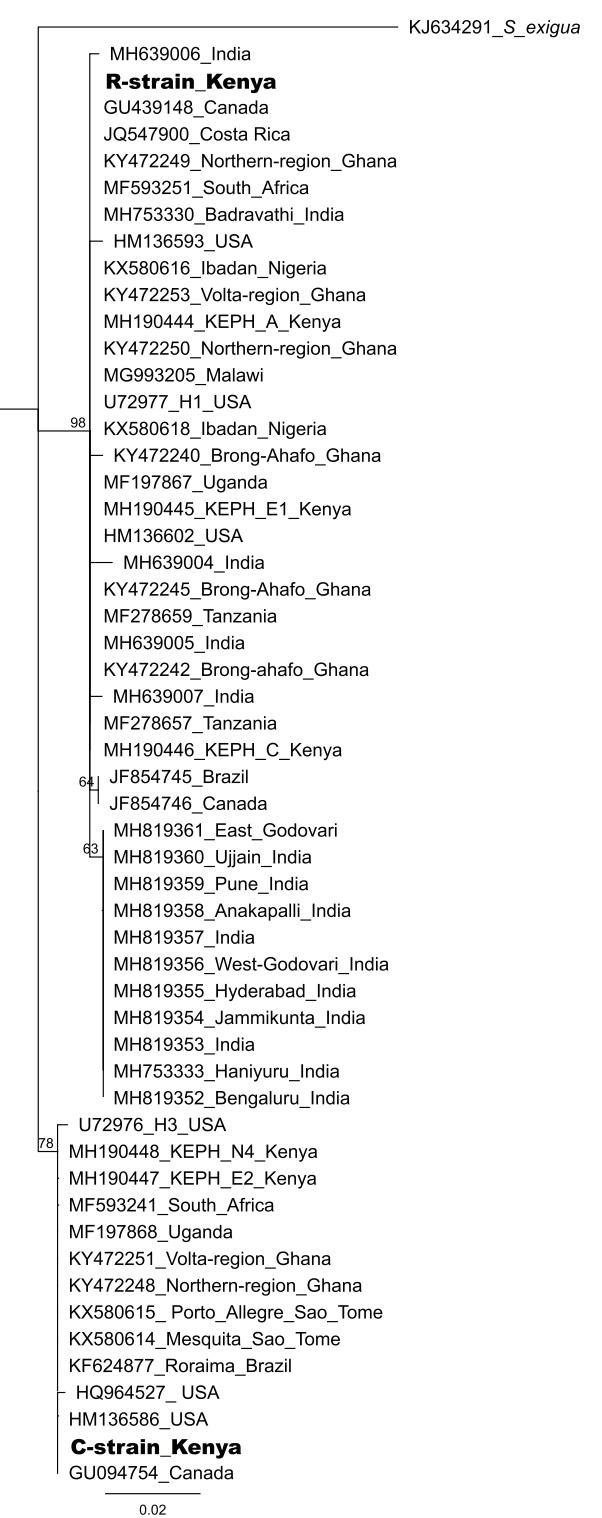

**Figure 9 Neighbor-joining tree based on mtCOI sequences of *S. frugiperda* from the GenBank and representative haplotypes from this study (in bold).** Bootstrap values are indicated above branches. Branches with bootstrap values less than 50 are collapsed. A sequence from *Spodoptera exigua* is included as an out-group. Sequences are labeled with their GenBank accession numbers, collection site where available and country of collection.

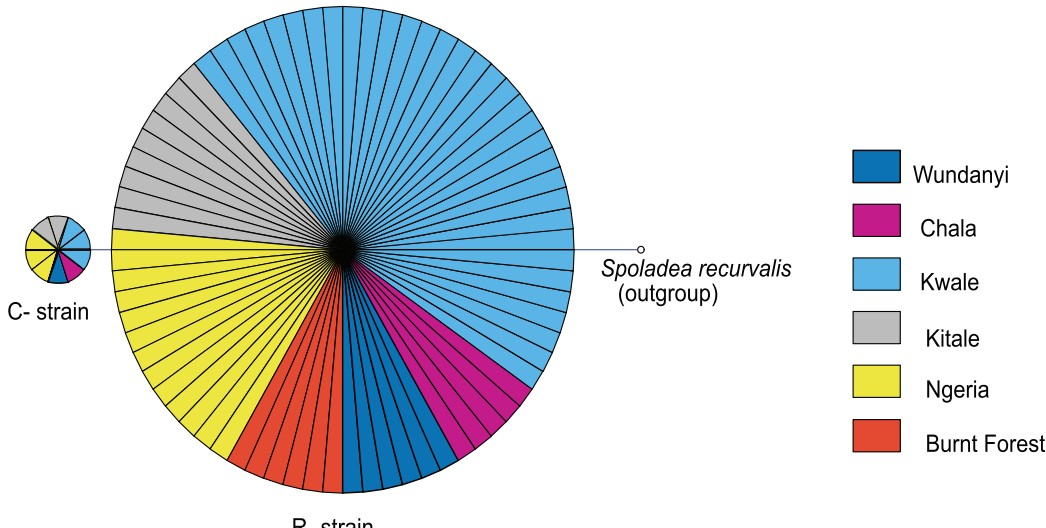

**Figure 10 Mitochondrial COI haplotype map of the *S. frugiperda* samples collected at various sites in Kenya.** Node size is proportional to number of samples and individual samples are represented as fractions of the nodes. A sequence of *Spoladea recurvalis* is included as an out-group. Sequences for all samples are accessible from GenBank using the accessions: MK492929–MK493010.

*S. frugiperda* samples collected at several sites in Kenya. Overall, 90% of the samples (*n* = 85) clustered as R-strain, whereas 10% (*n* = 9) clustered as C-strain. Proportions of the R-strain in populations at the different sites were 100% (*n* = 6) for Burnt Forest, 83% (*n* = 6) for Chala, 86% (*n* = 7) for Wundanyi, 82% (*n* = 11) for Kitale, 91% (*n* = 35) for Kwale and 82% (*n* = 17) for Ngeria (Fig. 10).

## DISCUSSION

We found that the gut bacterial communities of most *S. frugiperda* samples were dominated by Proteobacteria. This observation is similar to proportions reported in other phytophagous insects, in particular lepidopterans (*Belda et al., 2011*; *Xia et al., 2013, 2017*; *Landry et al., 2015*; *Ramya et al., 2016*; *Snyman et al., 2016*; *Strano et al., 2018*; *Chen et al., 2018*). Only three samples, two adult males from Kitale (Kitale-m2 and Kitale-m3) and one larva from Ngeria (Ngeria-l2) were dominated by Firmicutes. The four genera of bacteria, *Pseudomonas*, *Delftia*, *Enterococcus* and *Serratia*, that were recorded in this study have previously been isolated from *S. frugiperda* (*De Almeida et al., 2017*; *Acevedo et al., 2017*). Surprisingly, *Staphylococcus*, *Microbacterium*, *Arthrobacter* and *Leclercia* that were previously isolated from *S. frugiperda* in Brazil (*De Almeida et al., 2017*) were not found in any of the samples we profiled in Kenya.

Similarly, *Pantoea*, *Enterobacter*, *Raoultella* and *Klebsiella* previously identified in oral secretions of *S. frugiperda* in Pennsylvania, USA (*Acevedo et al., 2017*) were not found in the profiled Kenyan samples. Six of the detected bacterial genera: *Enterococcus*, *Pseudomonas*, *Chryseobacterium*, *Sphingobacterium*, *Ochrobactrum* and *Acinetobacter*, have been detected using a similar sequencing approach in both *S. frugiperda* as well as in the corn earworm *Helicoverpa zea* (*Jones et al., 2019*). Similarly, seven of the detected

bacterial genera: *Enterococcus*, *Pseudomonas*, *Comamonas*, *Stenotrophomonas*, *Eshcerichia-Shigella*, *Acinetobacter* and *Carnobacterium*, have been reported using a similar approach in the beet armyworm, *S. exigua* (*Gao et al., 2019*). This suggests that some bacterial genera often associate with lepidopteran insects, although it is difficult to define a core microbiota for such a diverse insect order. The OTUs classified as *Candidiatus hamiltonella* using the Silva database were further investigated and reclassified as *Pseudomonas*. *Candidiatus hamiltonella* has been recorded in whiteflies, psyllids and phloem-feeding relatives of the aphids (*Russell & Moran, 2005*) but not among lepidopteran insects.

We observed significant differences in OTU composition between larvae from different sites. This was most likely caused by complex biological and environmental factors in the diverse agro-ecological zones that were sampled. Diet is known to strongly influence the microbiome of lepidopterans (*Strano et al., 2018*; *Sittenfeld et al., 2002*; *Priya et al., 2012*; *Montagna et al., 2016*), however in this study all samples were collected from maize plants. Hence, the observed compositional differences are not likely to be caused solely by diet. It is interesting that many of the detected bacterial genera such as *Stenotrophomonas*, *Sphingobacterium*, *Serratia*, *Pseudomonas*, *Morganella*, *Enterococcus* and *Delftia* were found in both life stages, which suggests that gut bacterial community members are transmitted across developmental stages. Bacteria that are continually transmitted across developmental stages (and across generations) may evolve a closer, mutualistic relationship with their hosts (*Moran, 2006*). Future studies should investigate the effects of these microbes on host fitness and investigate the extent to which they are vertically transmitted from parents to offspring. In contrast, *Citrobacter* and *Sphingobacterium* were observed to be differentially abundant in larvae than in adults, a likely indicator that these two genera may be part of the fraction of bacterial communities that are lost during transition of *S. frugiperda* into the adult stage. *Lysinibacillus*, on the other hand, was more abundant in adults than in larvae and therefore could have an adult-specific function.

Notably, we identified *Serratia*, *Lysinibacillus* (formerly *Bacillus*) and *Pseudomonas*, species of which have been reported to have entomopathogenic properties (*Castagnola & Stock, 2014*). In addition, one sample had a high number of reads attributed to a relative of a fungal entomopathogen, *Metarhizium rileyi*, which previously has been isolated and tested for efficacy against *S. frugiperda* (*Maniania & Fargues, 1985*; *Mallapur et al., 2018*). However, there was no record of the use of any fungal biopesticides in any of the sampled sites. This detection could however be an indication that *S. frugiperda* might be refractory to the fungus, in which case such a fungus could be exploited in alternative control techniques such as in the delivery of dsRNA. It may be worthwhile to re-examine the pathogenicity of these microbes for *S. frugiperda* and to determine if they could be incorporated into biological pest management strategies (*Ruiu, 2015*).

Based on the mtCOI gene sequences, we observed two mtDNA haplotypes in Kenya (C- and R-strains), despite the fact that all of these insects were obtained from maize. These findings confirm that both haplotypes are present in Kenya, as has been demonstrated for other countries in Africa (*Rwomushana et al., 2018*). The majority of the

*S. frugiperda* samples collected were characterized as R-strain suggesting that this strain is dominant in *S. frugiperda* populations in Kenya. These observations are in agreement with a previous study (*Nagoshi et al., 2018*) that observed that C- and R-strains appear to have an East-West axis alignment in the African region, with Eastern Africa having progressively lower frequencies of the C-strain (*Nagoshi et al., 2018*). We noted that some variants of the R-strain have been reported in other places such as Ghana and India, but those variants were not detected in this study. It is interesting to note that in addition to an R-strain similar to the one detected in Kenya, a variant differing by a single nucleotide polymorphism in the sequenced region of the mtCOI gene has been recorded from various locations in India. This variant has however not been reported in Africa. It is therefore possible that the invasion into India may not have come directly from the African continent, or invasion could have included strains from Africa and elsewhere.

## CONCLUSIONS

We characterized the gut bacterial communities in *S. frugiperda* larvae and adult samples collected from several locations in Kenya, finding some important differences and similarities across samples and in relation to other studies on this species (*Acevedo et al., 2017*; *De Almeida et al., 2017*). Characterizing the gut microbial symbionts of this pest species in Africa can be seen as an important first step towards the development of novel, cost-effective symbiont and entomopathogen-based control strategies.

In addition, the population structure of this pest in Kenya was investigated. Understanding the population structure, dynamics and bio-ecology of invasive species is important for identifying their invasion patterns and for informing cropping systems especially where pest species compositions associate with different host plant usage.

## ACKNOWLEDGEMENTS

The authors acknowledge the technical support of Mr. Peter Malusi during sampling.

### Funding

Financial support for this research came from the following organizations and agencies: European Union Funded project "Integrated pest management" strategy to counter the threat of invasive fall armyworm to food security in Eastern Africa (FAW-IPM) (FOOD/2018/402-634); UK Aid from the UK Government; Swedish International Development Cooperation Agency (Sida); the Swiss Agency for Development and Cooperation (SDC); Federal Democratic Republic of Ethiopia; and the Kenyan Government. Financial assistance for sequencing at the Center for Integrated Genomics, Lausanne, was provided by the Global Health Institute of the École Polytechnique Fédérale de Lausanne. The funders had no role in study design, data collection and analysis, decision to publish, or preparation of the manuscript.

## Grant Disclosures

The following grant information was disclosed by the authors:

European Union: FOOD/2018/402-634.

UK Aid from the UK Government.

Swedish International Development Cooperation Agency (Sida).

Swiss Agency for Development and Cooperation (SDC).

Federal Democratic Republic of Ethiopia.

Kenyan Government.

Global Health Institute of the École Polytechnique Fédérale de Lausanne.

## Competing Interests

The authors declare that they have no competing interests.

## Author Contributions

- Joseph Gichuhi performed the experiments, analyzed the data, prepared figures and/or tables, authored or reviewed drafts of the paper, and approved the final draft.
- Subramanian Sevgan conceived and designed the experiments, authored or reviewed drafts of the paper, and approved the final draft.
- Fathiya Khamis conceived and designed the experiments, authored or reviewed drafts of the paper, and approved the final draft.
- Johnnie Van den Berg conceived and designed the experiments, authored or reviewed drafts of the paper, and approved the final draft.
- Hannalene du Plessis conceived and designed the experiments, authored or reviewed drafts of the paper, and approved the final draft.
- Sunday Ekesi conceived and designed the experiments, authored or reviewed drafts of the paper, and approved the final draft.
- Jeremy K. Herren conceived and designed the experiments, analyzed the data, authored or reviewed drafts of the paper, and approved the final draft.

## Data Availability

Gut bacteria reads are available in the Sequence Read Archive (SRA) BioProject: PRJNA521837.

Fall armyworm mtCOI sequences are available at GenBank: MK492929 to MK493010.

## Supplemental Information

Supplemental information for this article can be found online at http://dx.doi.org/10.7717/peerj.8701#supplemental-information.

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
