# Peer review of "Diversity of fall armyworm, Spodoptera frugiperda and their gut bacterial community in Kenya"

_PeerJ, doi:10.7717/peerj.8701_

## Round 0.1 · original submission · Major Revisions

Please review the comments from both reviewers and revise accordingly. Both have made suggestions to improve the manuscript, and reviewer 2 has indicated that more information is needed on how this information could be used in pest management. I believe that addressing these comments in a revision will provide a more thorough and detailed study, and I invite you to resubmit.

Reviewer 1 ·

Basic reporting

This paper general approach was to investigate the S. frugiperda microbiome ad its role in the insect’ success and adaptability to rice and maize cultivars in Kenya, Africa.


1. They compared the microbial composition, abundance and diversity of S. frugiperda larva and adult specimens collected from maize and rice from growing regions in Kenya.
2. They analyzed the bacterial diversity between larvae and adults.
3. They determined this insect population prevalence within maize and rice cultivars.

Authors identified mainly Proteobacteria and Firmicutes. They detected differences in bacterial diversity between larvae and adults, where some but not all prominent bacterial groups in larval stage are lost during metamorphosis, thus suggesting being transmitted.
They detected differences in bacterial diversity between larvae and adults, where some but not all prominent bacterial groups in larval stage are lost during metamorphosis, thus suggesting being transmitted. They detected several entomopathogenic bacterial clades and one entomopathogen fungus. DNA haplotyping of this insect population indicated higher S. frugiperda prevalence of the ‘rice’ strain.

Since microbiome plays an important role among insects’ prevalence and adaptability to different environments, this study is important. Based on this study results, I do recommend this manuscript for publication.

Experimental design

All experiments were well designed and performed in concordance to this study aims.

Main concern
Why the authors did not analyze larvae and adults from sorghum cultivars??

Validity of the findings

Data analyses from experiments were performed adequately to overcome this study purposes.

Additional comments

Suggested editing of this manuscript are:

Abstract

Where the insect were collected?
Ln 28. Here, authors stated “larval and adult specimens of S. frugiperda collected from four maize growing..”, whereas in Ln 126 (introduction section) they described collecting insects from rice in addition to maize. In the M&M section (ln 130) they stated “Spodoptera frugiperda larvae were collected from infested maize fields in Kenya.
Ln 29. Change “non-bacterial” by “fungic”

Introduction

Ln 52. Delete underlines before “et al., 2016”
Ln 53. Delete “Goergen et al., 2016”
Ln 53-54. Do authors need nine references to support the first paragraph? They should be able to select up to three.
Ln 65. Change “Further” by “Furthermore”
Ln 67-68. Do references really support the statement: S. frugiperda readily develop resistance to most chemical insecticides and transgenic cultivars?
Ln 69. The “Yu 1991” reference is out of place.
Ln 70. Change “In light of this,” by “In resume,”
Ln 93-96. Do authors need eight references to support the first paragraph? They should be able to select up to three.
Ln 126. Here, authors describe collecting insects form rice in addition to maize, whereas in the abstract (ln 28) they stated “larval and adult specimens of S. frugiperda collected from four maize growing..”

Material and Methods

Ln 130. S. frugiperda larvae were collected only from infested maize fields?

Results
Ln 189-190. Move paragraph “In addition, samples were collected from these 4 sites plus two additional sites for mtDNA haplotyping” to the Insect collection subsection of the Material and Methods section. Please specify if those samples were from rice cultivars.

Ln 192. Add “where” before “the median length”
Ln 204. Change “note” by “noticed”
Ln 210. Change “observed” by “detected”
Ln 2015. Change “fungi” by “fungus”


Discussion
Ln 266-267. Here the authors again state that all samples were collected from maize plants.
Ln 232. Delete “Other”
Ln 284. Change “non-bacterial” by “fungic”


Conclusions
Ln 310. “Symbiotic bacteria play a key role in the biology of insects.” is not a conclusion.

Reviewer 2 ·

Basic reporting

The manuscript contains genomic analyses conducted on specimens of Spodoptera frugiperda aimed at identifying and quantifying the main bacterial groups.
I believe that the work is not acceptable in the present state, because of some major concerns:
1) The introduction creates the expectation of an original study aimed at identifying potential microbiological control agents, but no observation is conducted on bacteria identified in the insect's microbioma to establish such potential. The objectives of the work should be clearer and in line with what has actually been done. This requires a significant revision of the introduction.
2) It is not clear which species have been identified. Does the work focus on bacterial taxa or information on species are available? Some species is mentioned in the discussion as a potential biocontrol agent. Please clarify.
3) Lack of comparison with the microbiome of a reference species, to make original this work referred to the Spodopter frugiperda
4) Fungi (eg. Metarhizium, etc.) are included in this work.; were they detected with bacterial 16S analyses. This has to be clarified.
5) Tha actual originality of this work should be clearly stated, based on the significance of the results.

Experimental design

Revise this section on the basis of previous comments.
Clarify methods of detection of fungi and bacteria.

Validity of the findings

Revise results and discussion on the basis of previous comments and according with the actual study objectives.

Additional comments

Because of major concerns I recommend to deeply revise this manuscript.

---

## Round 0.2 · Minor Revisions

I have made additional edits -- please see if you agree. Also, I have made a few comments.

---

## Round 0.3 · accepted · Accept

Thanks for your consideration of edits to improve the manuscript.